# Wu's Method Boosts Symbolic AI to Rival Silver Medalists and AlphaGeometry to Outperform Gold Medalists at IMO Geometry

**Shiven Sinha**[1][*]    **Ameya Prabhu**[2][*]    **Ponnurangam Kumaraguru**[1]
**Siddharth Bhat**[3][+]    **Matthias Bethge**[2][+]

[1]IIIT Hyderabad        [2] Tübingen AI Center, University of Tübingen        [3]University of Cambridge

## Abstract

Proving geometric theorems constitutes a hallmark of reasoning, combining intuitive, visual, and logical skills. This makes automated theorem proving of Olympiad-level geometry problems a milestone for human-level automated reasoning. AlphaGeometry, a neuro-symbolic model trained with 100M synthetic samples, solved 25 of 30 International Mathematical Olympiad (IMO) problems. It marked a major breakthrough compared to the reported baseline using Wu's method which solved only 10. Revisiting the IMO-AG-30 benchmark, we find that Wu's method is surprisingly strong and solves 15 problems, including some unsolved by other methods. This leads to two key findings: (i) Combining Wu's method with the classic synthetic methods of deductive databases and angle, ratio & distance chasing solves 21 out of 30 problems on a CPU-only laptop limited to 5 minutes per problem. Essentially, this classic method solves just 4 fewer problems than AlphaGeometry and establishes the first *fully symbolic* baseline that rivals the performance of IMO silver medalists. (ii) Wu's method even solves 2 of the 5 problems that AlphaGeometry failed on. Combining both, we set a new state-of-the-art for automated theorem proving on IMO-AG-30 solving 27 out of 30 problems – the first AI method which outperforms an IMO gold medalist.

## 1   Introduction

Automated theorem proving has been the long-term goal of developing computer programs that can match the conjecturing and proving capabilities demanded by mathematical research [10]. This field has recognized solving Olympiad-level geometry problems as a key milestone [2, 3], marking a frontier of computers to perform complex mathematical reasoning. The International Mathematical Olympiad (IMO) started in 1959 and hosts the most reputed theorem-proving competitions in the world that play an important role in identifying exceptional talents in problem solving. In fact, half of all Fields medalists participated in the IMO in their youth, and matching top human performances at the olympiad level has become a notable milestone of AI research.

Euclidean geometry is well suited to testing the reasoning skills of AI systems. It is finitely axiomatized [14] and many proof systems for Euclidean geometry have been proposed over the years which are amenable to automated theorem proving techniques [4, 5]. Furthermore, proof search can be guided by diagrammatic representations [12, 17], probabilistic verification [11, 21], and a vast array of possible deductions using human-designed heuristics for properties like angles, areas, and distances, methods affectionately called "*trig bashing*" and "*bary bashing*" [22, 23] by International Mathematical Olympiad (IMO) participants. In addition, this domain is challenging — specific proof systems need to be defined for specifying the problem, there is a shortage of data to train

---

[*]authors contributed equally, [+] equal advising

from, and problems typically contain ambiguities around degenerate cases [27, 19, 16] that are complex to resolve and have led to the humorous folklore that "*geometry problems never take care of degeneracies*".

Automated reasoning in geometry can be categorized into algebraic [26, 25, 15] and synthetic methods [12, 6, 20]. Recent focus has been on synthetic methods like Deductive Databases (DD) [6] that mimic human-like proving techniques and produce intelligible proofs by treating the problem of theorem proving as a step-by-step search problem using a set of geometry axioms. For instance, DD uses a fixed set of expert-curated geometric rules which are applied repeatedly to an initial geometric configuration. This is performed until the system reaches a fixpoint and no new facts can be deduced using the available rules. AlphaGeometry [24], a novel neuro-symbolic prover, represents a recent breakthrough advancement in this area. It adds additional rules to the prior work of DD to perform angle, ratio, and distance chasing (AR), and the resulting symbolic engine (DD+AR) is further enhanced using constructions suggested by a large language model (DD+AR+LLM-Constructions) trained on 100 million synthetic proofs. It has outclassed algebraic methods by solving 25 of 30 IMO problems, whereas the reported baseline based on Wu's method [26, 8] solved only ten [24].

Algebraic methods, such as Wu's method and the Gröbner basis method [15], transform geometric hypotheses into system of polynomials to verify conclusions. They offer powerful procedures that are proven to decide statements in broad classes of geometry [8, 15]. More precisely, Wu's method possesses the capability to address any problem for which the hypotheses and conclusion can be expressed using algebraic equations [7], while simultaneously generating non-degeneracy conditions automatically [27, 16]. This remarkable feature implies that Wu's method can handle problems not only in plane geometry but also in solid and higher-dimensional geometries, i.e. in areas where synthetic methods can be used only with great effort and additional considerations. [9].

Rather than indiscriminately tackling arbitrary problem instances, mathematicians concentrate their efforts on statements exhibiting specific properties that render them interesting, meaningful, and tractable within the broader context of mathematical inquiry [13]. In this work, we put the capabilities of Wu's method to the test on such structured problems and re-evaluate Wu's Method on the IMO-AG-30 benchmark introduced by Trinh et al. [24]. We find that it performs surprisingly strong, solving 15 problems, some of which are not solved by any of the other methods. This leads to two key findings:

- Combining Wu's method (Wu) with the classic synthetic methods of deductive databases (DD) and angle, ratio, and distance chasing (AR) solves 21 out of 30 methods by just using a CPU-only laptop with a time limit of 5 minutes per problem. Essentially, this classic method (Wu&DD+AR) solves just 4 problems less than AlphaGeometry and establishes the first *fully symbolic* baseline, strong enough to rival the performance of an IMO silver medalist.

- Wu's method even solves 2 of the 5 problems that AlphaGeometry (AG) failed to solve. Thus, by combining AlphaGeometry with Wu's method (Wu&AG) we set a new state-of-the-art for automated theorem proving on IMO-AG-30, solving 27 out of 30 problems, the first AI method which outperforms an IMO gold medalist.

## 2 Experiments & Results

### 2.1 Dataset

In January 2024, IMO-AG-30 was introduced as a new benchmark by Trinh et al. [24] to demonstrate the skill level of AlphaGeometry. IMO-AG-30 is based on geometry problems collected from the IMO competitions since 2000 and adapted to a narrower, specialized environment for classical geometry used in interactive graphical proof assistants, resulting in a test set of 30 classical geometry problems. The number of problems solved in this benchmark are related to the number of problems solved on average by IMO contestants. As indicated by the gray horizontal lines in Figure 1 (A), bronze, silver and gold medalists on average solved 19.3, 22.9 and 25.9 of these problems, and 15.2 represents the average over all contestants. The specific set of problems that have been collected for IMO-AG-30 are listed in the left column of the diagram in Figure 1 (B).

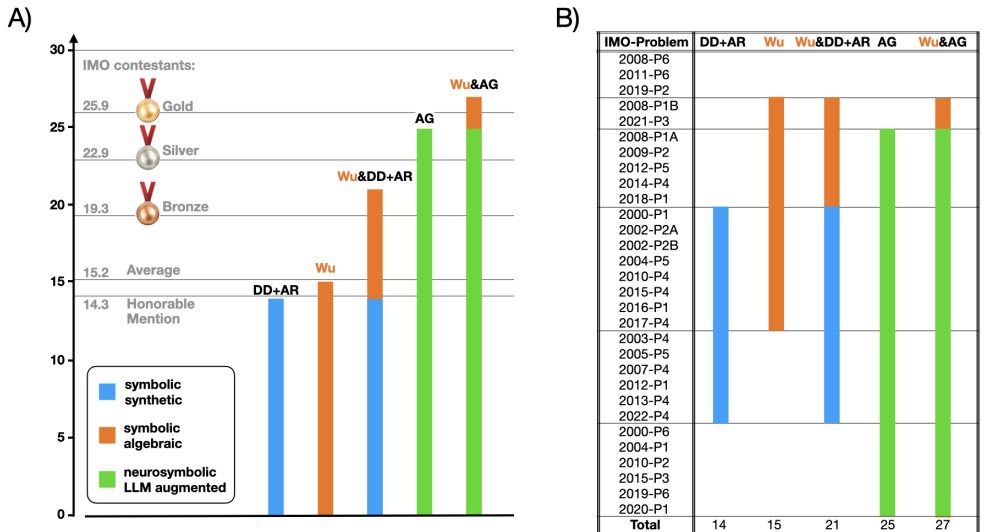

Figure 1: A) Performance across symbolic and LLM-Augmented methods on the IMO-AG-30 problem set, along with human performance. We set a strong baseline among symbolic systems at the standard of a silver medalist and outperform a gold medalist by a margin of one problem. B) Diagram showing how the different methods overlap or complement each other on the IMO-AG-30 problems.

## 2.2 Experimental Details

We evaluated performance using the IMO-AG-30 benchmark, with baselines and dataset all adopted from Trinh et al. [24]. We only re-implemented Wu's Method through the JGEX software [18, 28] by manual translation of the IMO-AG-30 problems into JGEX-compatible format[2]. We also successfully reproduced the DD+AR baseline, necessary for our final proposed method from the AlphaGeometry codebase. We manually verified that the hypothesis and conclusion equations generated by JGEX for several problems translated by us were indeed correct.

## 2.3 Results and Analysis

Our findings, are displayed in Figure 1 in combination with previous results from [24]. Figure 1 (A) compares the number of problems solved, and (B) shows which problems are solved by which method to visualize how the different methods overlap or complement each other. In Figure 1 (A), the performance levels of IMO contestants are indicated by gray horizontal lines, showing gold, silver, bronze, average, and honorable mention level. The performance levels of synthetic symbolic methods are displayed with blue bars and of LLM-augmented neurosymbolic methods are shown with green bars. Our own results obtained with Wu's method fall into the category of algebraic synthetic methods shown with orange bars. All results for synthetic symbolic methods (blue) or neurosymbolic LLM-augmented methods (green) are adopted from Trinh et al. [24].

Our combination of Wu's method with DD+AR sets a new symbolic baseline (Wu&DD+AR) that outperforms all traditional methods by a margin of 6 problems. It solves 21 of the IMO-AG-30 problems, matching the level of AlphaGeometry without fine-tuning (FT-9M only) shown in the Appendix (Figure 2). Wu's method achieves this performance with remarkably low computational requirements. On a laptop equipped with an AMD Ryzen 7 5800H processor and 16 GB of RAM, we were able to solve 14 out of 15 problems within 5 seconds. One problem (2015 P4) required 3 minutes. In our experiments, Wu's method either solves problems almost immediately or the laptop runs out of memory within 5 minutes. Remarkably, two of the fifteen problems we were able to solve with Wu's method (2021 P3, 2008 P1B) were among the five problems that were too difficult

---

[2]However, 4 out of 30 problems were untranslatable due to lack of appropriate constructions within the JGEX framework, hence our reported is out of 26 problems.

to solve for AlphaGeometry. Thus, by simple ensemble combination between Wu's method and AlphaGeometry, we obtain the new state-of-the-art solving 27 out of 30 problems on the IMO-AG-30 benchmark as visualized by the green/orange bar (Wu&AG) Figure 1.

## 3 Conclusion

Overall, our note highlights the potential of algebraic methods in automated geometric reasoning for solving International Mathematical Olympiad (IMO) geometry problems[3], raising the number of problems solved with Wu's method on IMO-AG-30 from ten to fifteen. Among those fifteen problems are several that are difficult for synthetic methods and their LLM-augmented versions that are currently most popular.

To the best of our knowledge, our symbolic baseline is the only symbolic baseline performing above the average IMO contestant and approaching the performance of an IMO silver medalist on geometry. Similarly, our combination of AlphaGeomtery with Wu's method is the first AI system to outperform a human gold-medalist at IMO geometry problems. This achievement illustrates the complementarity of algebraic and synthetic methods in this area (see Figure 1 B). The usefulness of algebraic approaches is most obvious from the two problems 2008 P1B and 2021 P3 which are currently solved by no automatic theorem prover other than Wu's method.

While algebraic methods have always been recognized for their theoretical guarantees, their usefulness has been previously questioned for being too slow and not human interpretable. Our observations indicate that on several problems Wu's Method performs more efficiently than previously recognized, and we advocate against dismissing it solely on the basis of its inability to produce human-readable proofs.

## 4 Limitations and Future Directions

Despite the theoretical promise, our results are a work-in-progress, currently hindered by the scarce availability of existing implementations, each with their significant inadequacies including limited constructions and suboptimal performance. We believe it might be feasible to outperform AlphaGeom-etry's proving capabilities through purely traditional methods and hope our note encourages improving current software for classical computational approaches in this area. Exploring improvements in the capabilities of other symbolic methods, including synthetic ones, in addition to extending the scope of geometry-specific languages and proof systems might be exciting directions to investigate.

Our exploration highlighting the complementary strengths of synthetic methods, which mimic human reasoning processes, and more abstract algebraic methods is motivated by the idea that the similarity to human reasoning and the generality of intelligence are distinct concepts, each with its own merits and applications. We believe that the strength of algebraic methods goes beyond solving Olympiad geometry problems, promising significant advancements in areas as varied as compiler verification and beyond. This potential underscores our belief in the necessity to broaden the scope of challenges addressed by automated theorem proving. The development of future benchmarks should strive for diversity and potentially open-ended testing. Embracing a wider array of problems will likely bring new insights on the usefulness, limitations, and interplay of neural and symbolic methods for general reasoning skills.

## Acknowledgements

The authors would like to thank (in alphabetic order): Shashwat Goel, Shyamgopal Karthik, Yash Sharma, Matthias Tangemann, Saujas Vaduguru for helpful feedback on the draft. MB acknowledges financial support via the Open Philanthropy Foundation funded by the Good Ventures Foundation.

---

[3]Peter Novotný similarly proved 11 of the 17 IMO Geometry problems from 1984–2003 using the Gröbner basis method, although only after manually adding non-degeneracy conditions [1] as referenced here.

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

# A   Detailed Comparisons

We compare with all human and automated methods on the IMO-AG-30 benchmark [24] in Figure 2. Our evaluation includes GPT4, Full-Angle method (FA), Gröbner Basis (Gröbner), Deductive Databases (DD), Deductive Databases combined with Algebraic Rules and enhancements with GPT-4 for construction suggestions (DD+AR+GPT4). Additionally, we examined different configurations of the AlphaGeometry model: one only pretrained on 100 million samples (PT-100M) and other only finetuned on 9 million constructions (FT-9M). Note that we construct the Wu&DD+AR baseline by simply parallelly running both Wu's and DD+AR methods and stopping when either method solves the problem. Similarly, we construct the Wu&AlphaGeometry baseline. We see that our Wu&DD+AR baseline matches AG (FT-9M) baseline while Wu's method alone matches the best DD+AR+GPT4 algorithm.

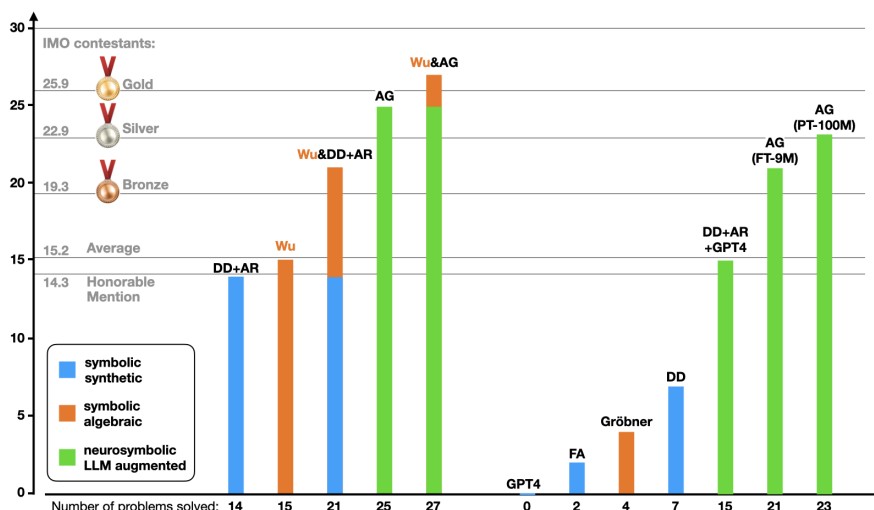

Figure 2: **Extended version of Figure 1A:** Performance across symbolic and LLM-Augmented methods on the IMO-AG-30 problem set, along with human performance. The performance of additional models adopted from Table 1 in [24] are shown on the right.

# B   Illustrations: 2008 P1B and 2021 P3

We provide illustrations of the solutions of Wu's method for the two problems AlphaGeometry could not solve to allow for additional scrutiny without having to reproduce the same on the JGEX solver.

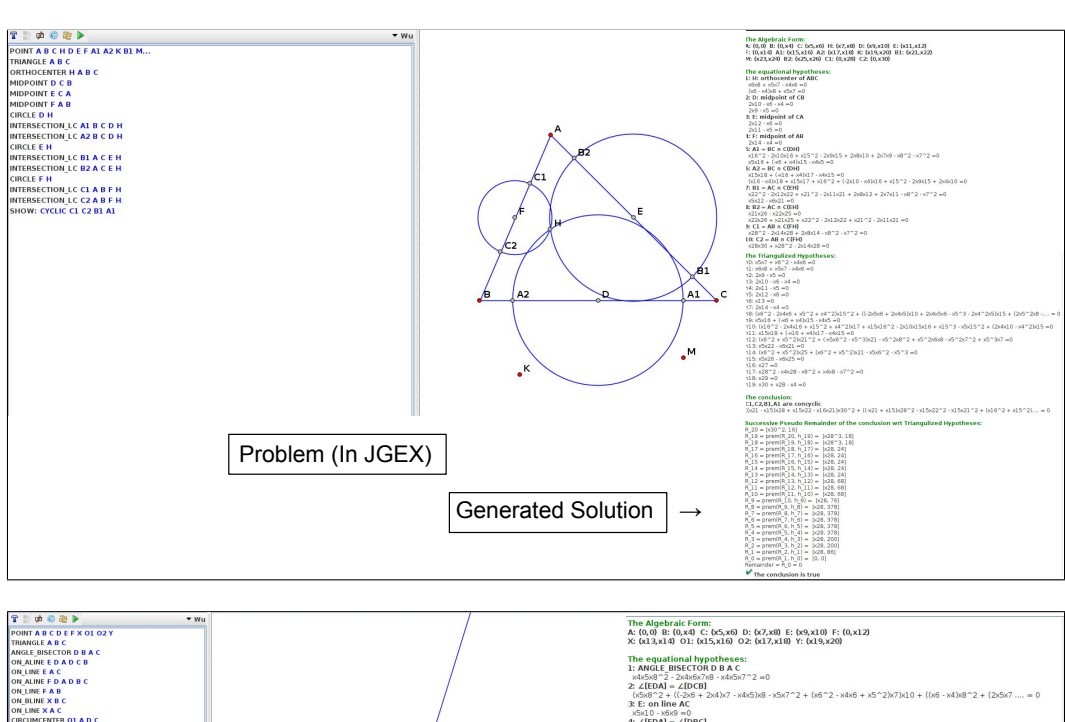

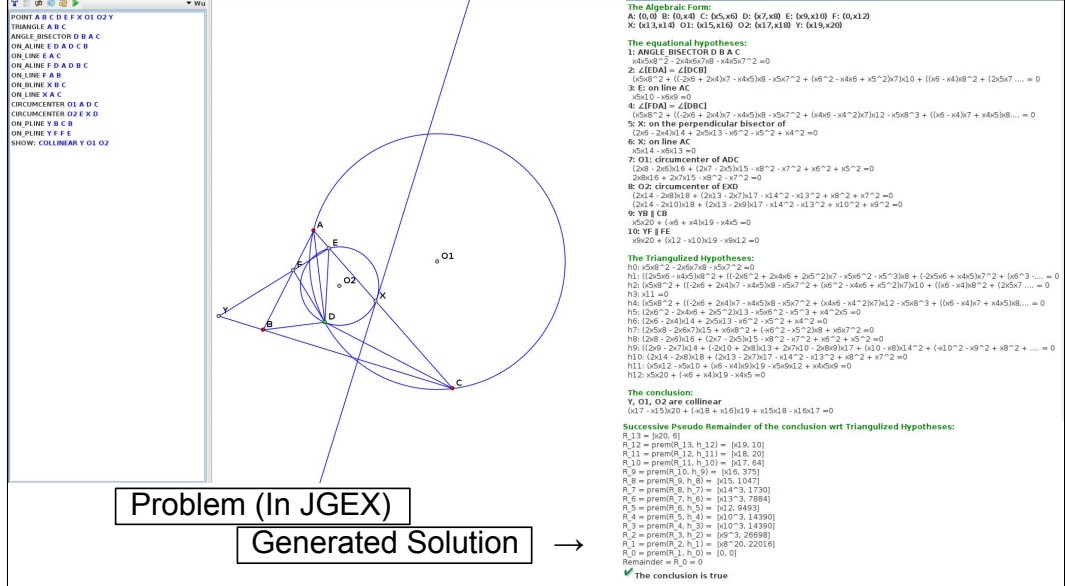

Figure 3: **Problem 2008-P1B JGEX (Above) and 2021-P3 (Below) with Input (Left) and Generated Solution (Right) for Wu's method.** This illustration can be reproduced by opening the .gex files provided alongside on the HuggingFace repository and pressing Run.

