# OpenReview forum: "Wu’s Method Boosts Symbolic AI to Rival Silver Medalists and AlphaGeometry to Outperform Gold Medalists at IMO Geometry"
_NeurIPS.cc/2024/Workshop/MATH-AI — MATH-AI 24_

### Official Review · Reviewer_eUix · 2024-10-05

**Rating:** 3
**Confidence:** 5

**Review:**

This paper demonstrates that Wu's method can solve 2 problems in IMO-AG-30 that AlphaGeometry cannot, and 7 that DD+AR, AlphaGeometry's deductive engine, cannot. It does not offer any technical contributions.

---

### Official Review · Reviewer_hrQc · 2024-10-07
**This paper emphasizes Wu’s method’s capability in the IMO-AG-30 benchmark.**

**Rating:** 6
**Confidence:** 4

**Review:**

This paper emphasizes Wu’s method’s capability in the IMO-AG-30 benchmark with the results that, combining Wu’s method with DD+AR solves 21 out of 30 problems on a CPU-only laptop limited to 5 minutes per problem. Moreover, Wu’s method solves 2 of the 5 problems that AlphaGeometry failed on.

Pros:
1. Exellent experimental results on re-discovering that algebraic methods, such as Wu’s method and the Gröbner basis method, perform very well on the challenging IMO-AG-30 benchmark.

Cons:
1. What are the technical contributions and novelty introduced in this paper?

2. More diverse problems should be collected to test algebraic methods and alphageometry, not limited on IMO-AG-30.

---

### Official Review · Reviewer_2mF6 · 2024-10-07
**Review of Wu&AG**

**Rating:** 9
**Confidence:** 3

**Review:**

This work revisits the capabilities of Wu's method in automated theorem proving for Olympiad-level geometry problems, particularly on the IMO-AG-30 benchmark. The authors report that Wu's method solves 15 out of 30 problems—an improvement over the previously reported 10. By combining Wu's method with classic synthetic methods like DD and AR, they achieve solutions for 21 problems using minimal computational resources. Furthermore, integrating Wu's method with AlphaGeometry leads to solving 27 out of 30 problems, surpassing the average performance of IMO gold medalists. The results from this work are impressive and interesting for the audience of MATHAI workshop.

I am wondering what the authors think about the reason behind different performances in AG and Wu's method. The paper lacks a detailed comparative analysis of why Wu's method succeeds where AlphaGeometry fails. That analysis could deepen our understanding of why AG was not successful in solving those two questions that Wu's method was. Furthermore, as the authors mentioned, they were not able to translate 4 problems in JGEX format, which is concerning regarding the generalizability of their work beyond the existing 30 problems.

Overall, I liked the idea behind this work and their analyses are significant and can be a great contribution to the workshop.

---

### Decision · Program_Chairs · 2024-10-09

Accept